# Changes in the Phenotype and Metabolism of Peritoneal Macrophages in Mucin-2 Knockout Mice and Partial Restoration of Their Functions In Vitro After L-Fucose Treatment

**DOI:** 10.3390/ijms26010013

**Published:** 2024-12-24

**Authors:** Elena L. Arzhanova, Yulia Makusheva, Elena G. Pershina, Snezhanna S. Medvedeva, Ekaterina A. Litvinova

**Affiliations:** 1Faculty of Natural Sciences, Novosibirsk State University, 630090 Novosibirsk, Russia; elena.arzhanova@hotmail.com (E.L.A.); pelena@bionet.nsc.ru (E.G.P.); 2Scientific Research Institute of Neurosciences and Medicine, 630117 Novosibirsk, Russia; 3Physical Engineering Faculty, Novosibirsk State Technical University, 630073 Novosibirsk, Russia; ys.makusheva@gmail.com; 4Federal Research Center Institute of Cytology and Genetics, Siberian Branch of the Russian Academy of Sciences, 630090 Novosibirsk, Russia; 5Institute of Molecular and Cellular Biology, 630090 Novosibirsk, Russia

**Keywords:** macrophages, mucin-2, fucose, inflammatory bowel disease

## Abstract

In the development of inflammatory bowel disease (IBD), peritoneal macrophages contribute to the resident intestinal macrophage pool. Previous studies have demonstrated that oral administration of L-fucose exerts an immunomodulatory effect and repolarizes the peritoneal macrophages in vivo in mice. In this study, we analyzed the phenotype and metabolic profile of the peritoneal macrophages from *Muc2^−/−^* mice, as well as the effect of L-fucose on the metabolic and morphological characteristics of these macrophages in vitro. The investigation utilized flow cytometry, quantitative PCR (qPCR), measurement of the intracellular ATP and Ca^2+^ concentrations, an analysis of mitochondrial respiration and membrane potential, and transmission electron microscopy (TEM) for ultrastructural evaluations. The *Muc2^−/−^* mice exhibited lower intracellular ATP and Ca^2+^ levels in their peritoneal macrophages, a higher percentage of stellate macrophages, and an increased oxygen consumption rate (OCR), combined with a higher percentage of mitochondria displaying an abnormal ultrastructure. Additionally, there was a five-fold increase in condensed mitochondria compared to their level in C57BL/6 mice. The number of CD209^+^ peritoneal macrophages was reduced three-fold, while the number of M1-like cells increased two-fold in the *Muc2^−/−^* mice. L-fucose treatment enhanced ATP production and reduced the expression of the *Parp1*, *Mt-Nd2*, and *Mt-Nd6* genes, which may suggest a reduction in pro-inflammatory factor production and a shift in the differentiation of peritoneal macrophages towards the M2 phenotype.

## 1. Introduction

IBD is a group of diseases associated with damage to the intestinal mucosa and the development of inflammation. Genetic predisposition, diet, immunity, the gut microbiota, and environmental factors play a role in the pathogenesis of this disease [1]. The development of inflammation leads to dysfunction of the intestinal mucosa, which results in malabsorption of nutrients and further development of extraintestinal manifestations. In recent years, the number of cases of IBD in children and young people has increased [1]. Antibiotics, aminosalicylates, corticosteroids, and immunosuppressants are prescribed for the treatment of IBD, but new methods are constantly being sought [2].

One of the most commonly used models of IBD is *Muc2^−/−^* mice, which spontaneously develop ulcerative colitis at the age of 2 to 20 weeks, often without symptoms [3,4]. Disruption of the mucin barrier, which normally separates the intestinal microbiota and the epithelial cells, leads to the development of inflammation. This is mainly associated with an increase in the production of antimicrobial factors such as resistin-like molecule β (Relmβ) and inducible nitric oxide synthase (iNOS) [3]. However, there are no differences in the apoptotic cells in the intestines of *Muc2^−/−^* and wild-type mice [5]. In addition, the gut microbiota of *Muc2^−/−^* mice was similar to that of patients with ulcerative colitis [3]. Depletion of the intestinal mucous layer promotes the activation of the immune system towards protection from pathogens and the prevention of hyperinflammation. These changes are associated with the activation of antigen presenting cells (APCs), in particular tissue macrophages [4].

Fucose is a monosaccharide found in 7.2% of 3299 mammalian oligosaccharides [6]. Mucin-2, a protein forming the mucus layer in the intestines, is a glycosylated protein and contains L-fucose in the terminal position. The absence of mucin-2 or the decreased function of fucosyltransferase (*Fut2*) can lead to microbiome changes [4]. Polymorphisms in the *Muc2* and *Fut2* genes lead to severe intestinal inflammation in humans [7,8]. Thus, not only a deficiency of mucin-2 but also the absence of L-fucose in the composition of mucin-2 can be important within the development of IBD.

There is now increasing evidence that L-fucose itself can affect the bacterial composition, as well as the inflammatory state, in the intestines. One recent study showed that L-fucose caused changes in the microbiome that activated the epithelial cells [9]. In addition, L-fucose may be a prebiotic for beneficial bacteria [10] which reduces inflammation in IBD [10]. It was shown that L-fucose has an effect on inflammation in *Muc2^−/−^* mice in vivo [11,12]. Furthermore, L-fucose treatment decreased macrophage M1 polarization via the inhibition of the NF-κB pathway in a DSS-induced colitis model [13]. L-fucose affects the polarization of the macrophages to the M1 or M2 type during pregnancy in C57BL/c mice [10]. Moreover, mucin-2-associated glycans activated anti-inflammatory dendritic cells through the β-catenin pathway [14]. Also, the metabolic rate of the peritoneal macrophages in *Muc2^−/−^* mice differs from that in cells from wild-type mice [15].

The polarization of macrophages also affects their type of respiration. That is, M1 macrophages use glycolysis to produce ATP, while M0 or M2 macrophages accumulate ATP and NADH through oxidative phosphorylation [16]. The causes of different types of respiration in macrophages are associated with the accumulation of reactive oxygen species (ROS) and mitochondrial dysfunction. The metabolic processes of a cell depend on the amount of NAD^+^, which is a cofactor for the key enzymes of glycolysis; oxidative phosphorylation; and the tricarboxylic acid cycle. NAD^+^ is degraded during catalytic processes by poly-ADP-ribose polymerases (PARPs), NAD-dependent deacetylases (SIRTUINS), and NADases such as CD38 [17]. NAD^+^ deficiency has been documented in age-associated chronic inflammation and correlates with increased CD38 expression. It was shown that low levels of intracellular NAD^+^ activated inflammation, associated with the release of IL-1β, and increased the surface expression of inflammatory markers, combined with defective phagocytosis [18]. NADases (such as CD38) control the release of IL-6 and IL-12, and inhibition of CD38 decreases lactate production, indicating a defect in glycolysis. Current evidence supports the idea that CD38-mediated NAD^+^ depletion contributes to inflammation [19]. Since poly(ADP-ribose) (PAR) can be synthesized by either CD38 or poly-ADP-ribose polymerase 1 (PARP-1), this may allow for the regulation of immune cell responses [20].

We hypothesized that an increase in the amount of free L-fucose interacting with the peritoneal macrophages in *Muc2^−/−^* mice may affect the inflammatory process and reduce the aggressiveness of these macrophages. There is a DC-SIGN receptor (CD209) on the macrophages that interacts with L-fucose [21], and activation of this receptor reduces the number of macrophages polarizing along the M1 pathway [16]. Similar data were shown in work by our laboratory [12]. However, it remains unclear whether this effect is the result of a direct interaction between L-fucose and the macrophages or whether it is mediated through the microbiota, where L-fucose may be a prebiotic.

To our knowledge, the absence of mucin-2 activates different pathways of peritoneal and tissue macrophage polarization, such as the β-catenin pathway, the NF-κB pathway, or the pathway downstream of the toll-like receptor (TLR). That is, we wanted to understand how L-fucose could change the phenotype of the peritoneal macrophages derived from C57BL/6 and *Muc2^−/−^* mice in terms of their cellular markers, shape, and type of respiration. The mitochondrial structures of the macrophages in the colon were analyzed using transmission electron microscopy, and the percentage of mitochondria with a high membrane potential in the peritoneal macrophages in C57BL/6 and *Muc2^−/−^* mice was assessed. The effect of L-fucose on the levels of ATP, Ca^2+^ ions, NADases (CD38), enzymes included in the β-catenin pathway (mitochondrially encoded NADH dehydrogenase 2 (MT-ND2), mitochondrially encoded NADH dehydrogenase 6 (MT-ND6), and leukotriene C4 synthase (LTC4S)) and the NF-κB pathway (IL-1), and enzymes responsible for the metabolism of poly(ADP-ribose) from NAD^+^ (poly(ADP-ribose) glycohydrolase (PARG) and PARP1) was evaluated in vitro.

## 2. Results

### 2.1. Macrophages in Muc2^−/−^ Mice Demonstrate M1-like-Type Characteristics

Firstly, the polarization of the peritoneal macrophages to the M1 and M2 types was determined using flow cytometry using the surface markers CD80, CD86, CD209, and CD206. In mice with a null mutation of the *Muc2* gene, the percentage of M2-like macrophages (CD209^+^) was three times lower than that in the wild-type mice (one-way PERMANOVA F(1,18) = 14.34; *p* = 0.003; Figure 1A), but the other marker of M2-like macrophages (CD206^+^) was similar in both groups (one-way PERMANOVA F(1,8) = 1.26 ND; Figure 1C). On the contrary, the percentage of M1-like macrophages (CD80^+^) in the *Muc2^−/−^* mice was two times higher compared to that in the C57BL/6 mice (one-way PERMANOVA F(1,18) = 27.04; *p* < 0.001; Figure 1B), and the other M1-like macrophage marker (CD86^+^) was also higher in the *Muc2^−/−^* mice (one-way PERMANOVA F(1,8) = 9.33; *p* < 0.01; Figure 1D). We also observed 47,93 ± 18.82% double-stained cells, with no difference between genotypes. The IHC analysis of the colon tissue showed an elevation in the resident macrophages in the *Muc2^−/−^* mice (Appendix A). The number of macrophages in the lamina propria of the C57BL/6 mice was too low to show any statistics. Analysis of the resident macrophages using CD80 and CD209 staining was conducted only for the *Muc2^−/−^* mice and showed the presence of single- and double-stained macrophages, similar to the results of the peritoneal macrophage analysis (Appendix A).

Since macrophages polarized according to different types have different shapes [21], we stained the samples with β-actin and F4/80 antibodies, assessed the shape of the cells using a fluorescent microscope, and calculated the percentage of spindle-shaped, star-shaped, and round cells (Figure 1E). The percentage of spindle-shaped and round cells was lower, but not significantly, in the *Muc2^−/−^* mice compared to the C57BL/6 mice (one-way PERMANOVA F(1,18) = 2.1 ND and F(1,18) = 2.2 ND, respectively; Figure 1C). The percentage of star-shaped cells was significantly higher in the *Muc2^−/−^* mice compared to the WT mice (one-way PERMANOVA F(1,18) = 9.8; *p* = 0.005; Figure 1C). To sum up, *Muc2*-deficient mice have more M1-like macrophages, and most of them are star-shaped.

It is known that M1-activated macrophages generate ATP through glycolysis, which requires more oxygen consumption in comparison to oxidative phosphorylation, which is used by M0- and M2-polarized macrophages [22]. We analyzed the oxygen consumption rate (OCR) in the peritoneal macrophages using Seahorse technology (Agilent). It was found that the OCR was significantly higher in the macrophages from the *Muc2^−/−^* mice as compared to those from the control mice (one-way PERMANOVA F(1,13) = 7.6; *p* = 0.02; Figure 1F). We found that the *Muc2*-derived peritoneal macrophages exhibited an increased OCR, which indicates severe defects in mitochondrial metabolism. At the same time, the macrophages from the C57BL/6 mice also demonstrated an increased OCR in response to FCCP but not as high an increase as that in the *Muc2^−/−^* mice’s macrophages (one-way PERMANOVA F(1,13) = 8.5; *p* = 0.02; Figure 1D,F). At the same time, ATP inhibition using oligomycin and the termination of mitochondrial respiration using an antimycin A/rotenone complex did not show any difference between the macrophages from the *Muc2^−/−^* and C57BL/6 mice (one-way PERMANOVA F(1,13) = 4.4 ND and F(1,13) = 3.6 ND; Figure 1F). Interestingly, only ATP-linked respiration but not proton leak was significantly increased in the *Muc2^−/−^* mice-derived macrophages compared to the C57BL mice’s macrophages (one-way PERMANOVA F(1,13) = 8.6 (*p* = 0.017) and F(1,13) = 1.7 ND, respectively; Figure 1E,H). Enhanced maximal and basal respiration led to an increase in the reserve capacity of the *Muc2^−/−^* mice-derived macrophages (one-way PERMANOVA F(1,13) = 6.4; *p* = 0.03; Figure 1H). Furthermore, basal glycolysis and glycolytic capacity were significantly higher in the *Muc2^−/−^* mice-derived macrophages compared to the C57BL/6 mice’s macrophages (one-way PERMANOVA F(1,13) = 14.44 (*p* < 0.001); F(1,13) = 15.28 (*p* < 0.001); Figure 1H).

Extracellular acidification of the medium (ECAR) is an indicator of the intensity of cellular respiration. Thus, the level of extracellular acidification of the medium during baseline respiration by the peritoneal macrophages, as well as after their exposure to oligomycin, was higher in the *Muc2^−/−^* mice compared to the C57Bl/6 mice (one-way PERMANOVA F(1,13) = 6.2 and PERMANOVA F(1,13) = 19.05, respectively; *p* < 0.01; Figure 1I). This means that levels of basal glycolysis and the glycolic capacity are higher in *Muc2^−/−^* compared to C57Bl/6 mice (one-way PERMANOVA F(1,13) = 6.2 and PERMANOVA F(1,13) = 19.05, respectively; *p* < 0.01; Figure 1H).

### 2.2. Mucin-2 Deficiency Affects the Mitochondrial Ultrastructure of the Peritoneal Macrophages and Macrophages in the Lamina Propria, as Well as the Mitochondrial Membrane Potential of the Peritoneal Macrophages

Using TEM, we discovered signs of inflammation in the colons of the *Muc2^−/−^* mice. The wall was thickened, and macrophages, lymphocytes, and eosinophils were plentiful in the lamina propria on the colonic sections. In the cytoplasm of the macrophages, an abundance of lysosomes and phagolysosomes was observed. The surface density of the mitochondria of the *Muc2^−/−^* mice (number per 1 µm^2^) did not change significantly compared to that in the control strain (Figure 2A,D). A large number of the *Muc2^−/−^* mice’s mitochondria had structural defects (Figure 2B,C). Even though the macrophages from the C57BL/6 mice also had mitochondria with an aberrant structure (Figure 2B), the average proportion of empty space in the matrix of the abnormal mitochondria was almost four times higher in the *Muc2^−/−^* mice than in the control mice (*p* < 0.001; Student’s *t*-test; Figure 2E), i.e., the severity of structural disorders was more prominent. Some mitochondria in the *Muc2^−/−^* mice had “empty” spaces occupying almost the entire area of the mitochondrion (Figure 2B,C).

The mean area of the mitochondrial section of the macrophages in the colon in the *Muc2^−/−^* mice was higher than that in the control mice (0.24 ± 0.02 µm^2^ and 0.18 ± 0.01 µm^2^, respectively; *p* < 0.05; Student’s *t*-test). The shape of the mitochondria was rounder. Though the length of mitochondria was comparable to that in the control, the width was higher, by 24.05% (*p* < 0.001, Student’s *t*-test). In addition to the above-mentioned presence of “empty” spaces in the matrix, the alteration in the mitochondria’s shape allows us to suggest edema of these organelles.

The surface density of the cristae in the mitochondrial matrix can be used to indirectly evaluate the level of oxidative phosphorylation. This parameter was almost identical in both groups (43.26 ± 1.96 and 43.32 ± 2.23 in the mutant and control animals, respectively), and the average length of the cristae was also similar; however, the average number of cristae per mitochondrion was 24.5% higher in the mutant mice (*p* < 0.05; Student’s *t*-test; Figure 2F). This phenomenon may have been caused by several factors. First, in swollen mitochondria, the cristae were displaced to undamaged areas. Second, the macrophage mitochondria in the *Muc2^−/−^* mice had dense arrangement of cristae, i.e., they were in a condensed state [23]. In the control group, the number of these condensed mitochondria was 5.71 ± 9.90%, while in the *Muc2^−/−^* mice, it was 21.67 ± 15.25%.

It should be noted that mitochondrial disarrangements were not unique to the macrophages. Empty spaces and other hallmarks of mitochondrial degradation were also observed in other mucosal cells: enterocytes, fibroblasts, lymphocytes, plasma cells, eosinophils, etc. (Appendix A). Previously, we noted similar structural impairments in enterocytes from *Muc2^−/−^* mice’s colons [15].

In the peritoneal macrophages, we did not observe such serious pathological changes as in those in the macrophages of the lamina propria. The cells were predominantly round in shape and had well-developed filopodia and lamellipodia. The nuclei were bean-shaped, with curved envelopes. In some cells, the perinuclear space was slightly dilated, which may have indicated the onset of apoptosis (Figure 3A). In the cytoplasm, there were a large number of lysosomes, as well as phagosomes and phagolysosomes. However, some changes in the morphology of the cells and their mitochondria were still marked. The cells from the mice from the *Muc2^−/−^* group were slightly smaller in size than those in the control group (36.9 ± 1.4 and 41.7 ± 2.6 µm^2^, respectively), but this difference was not significant (*p* = 0.08), but the average cytoplasmic area per section decreased by 25.6% (*p* = 0.0003; Student’s *t*-test; Figure 3B). The ultrastructure of the peritoneal macrophages’ mitochondria differed from the ultrastructure in those from macrophages in the lamina propria. The matrix looked denser, and we did not observe any “empties” or membrane disruptions in the control group nor in the *Muc2^−/−^* mice. The peritoneal macrophages from the *Muc2^−/−^* mice had larger mitochondria: the average area of this organelle was higher by 17.3% (*p* = 0.007; Student’s *t*-test; Figure 3C). At the same time, the numerical density of mitochondria in the *Muc2^−/−^* mice was similar to that the control group (0.66 ± 0.03 µm^−1^ and 0.59 ± 0.03 µm^−1^, respectively). The roundness of the mitochondria was similar in both groups. The average length and numerical density of the cristae did not differ between groups either.

In the case of the peritoneal macrophages, we were able to determine the proportion of condensed mitochondria in a section of a whole cell, rather than its fragments, as in the case of tissue macrophages. Therefore, such measurements were more relevant. In the control group, the proportion of condensed mitochondria was 5.37 ± 1.57%, and in the *Muc2^−/−^* group, it was 25.47 ± 2.65% (*p* = 6.35E-09; Student’s *t*-test; Figure 3D).

To understand whether mitochondrial function is impaired in the peritoneal macrophages of *Muc2^−/−^* mice, we assessed the mitochondrial membrane potential using JC-1 staining, with further analysis using a FACS assay (Figure 3E). It was shown that the percent of peritoneal macrophages with a high mitochondrial membrane potential (ΔѰm) was decreased in the *Muc2^−/−^* mice compared to that in the C57BL/6 mice (*p* < 0.01; Student’s *t*-test; Figure 3F). However, the percent of cells with a low mitochondrial membrane potential was no higher than 4%, and *Muc2* deficiency did not affect this (Figure 3F).

### 2.3. L-Fucose Affects Peritoneal Macrophage Metabolism

Since L-fucose can have an effect on reducing inflammatory processes in mice, not only through direct interactions with the immune cells but also indirectly through interaction with the microbiota, we investigated the effect of L-fucose on a primary culture of peritoneal macrophages to understand the direct effect of L-fucose. To determine the role of L-fucose on the change in the amount of pro- and anti-inflammatory cytokines, peritoneal macrophages from two lines of mice, *Muc2^−/−^* and C57BL/6, were cultured for 24 h with or without L-fucose in medium. The concentration of cytokines in the supernatant was determined using a multiplex assay. The levels of IL-1α (Mann–Whitney test: Z = 13.01; *p* < 0.05), IL-6 (Z = 4.95; *p* < 0.05), MIP-1α (Z = 6.49; *p* < 0.05), MIP-1β (Z = 8.62; *p* < 0.05) (MIP—macrophage inflammatory protein), and KC (U = 0.00; Z = 6.31; *p* < 0.05) (KC—keratinocyte-derived chemokine neutrophil chemoattractant) in the supernatant of the peritoneal macrophages from the *Muc2^−/−^* mice were significantly higher compared to those in the C57BL/6 mice (Figure 4A). The addition of 0.1% L-fucose increased the IL-1α levels only in the culture of the peritoneal macrophages derived from the C57BL/6 mice (Mann–Whitney test: Z = 12.89; *p* < 0.05) and had no significant effect on the macrophages from the *Muc2^−/−^* mice. However, the addition of 0.1% L-fucose eliminated the difference between the genotypes (Figure 4A).

Next, the metabolic features of the peritoneal macrophages from the C57BL/6 and *Muc2^−/−^* mice, including before and after their exposure to L-fucose, were analyzed. Firstly, *Muc2* deficiency itself had a significant effect on the ATP levels in the peritoneal macrophages. Thus, the levels in *Muc2^−/−^* mice-derived macrophages were significantly lower compared to those in the macrophages from the C57BL/6 mice (two-way PERMANOVA test F(1,44) = 4.20; *p* = 0.04; Figure 4B). The addition of 0.1% L-fucose to the medium had no significant effect on the ATP levels in the mice from both genotypes (two-way PERMANOVA test F(1,44) = 1.19 *ND*; Figure 4B).

Ca^2+^ ions are known to be involved in macrophage polarization and phagosome maturation. In our experiment, after incubation of the macrophages in vitro, the content of Ca^2+^ in the peritoneal macrophages was significantly lower in those from the *Muc2^−/−^* mice compared to those from the C57BL/6 mice (two-way PERMANOVA test F(1,44) = 6.29; *p* = 0.02; Figure 4C). Furthermore, the addition of 0.1% L-fucose to the growth medium significantly reduced the amount of Ca^2+^ ions in the peritoneal macrophages from the *Muc2^−/−^* mice only (two-way PERMANOVA test F(1,44) = 4.37; *p* = 0.05; Figure 4C).

CD38 is an NADase, and its increased expression correlates with NAD^+^ deficiency and the development of inflammation. It can act as either an enzyme inside the cell or a cellular marker on the cell surface. The percentage of peritoneal macrophages with CD38 on their cell surface was significantly higher in the *Muc2^−/−^* mice compared to the control group of C57BL/6 mice (two-way PERMANOVA test F(1,20) = 17.58; *p* < 0.0001; Figure 4D). At the same time, the addition of 0.1% of L-fucose during incubation significantly increased the percentage of CD38 on the cell surface for the *Muc2^−/−^* mice’s macrophages (two-way PERMANOVA test F(1,20) = 6.55; *p* = 0.015; Figure 4D). The percentage of peritoneal macrophages containing intracellular CD38 was significantly higher in the *Muc2^−/−^* mice compared to the C57BL/6 mice (two-way PERMANOVA test F(1,17) = 16.26; *p* = 0.002; Figure 4E). The addition of L-fucose significantly enhanced the percentage of intracellular CD38 expression for the *Muc2^−/−^* mice’s macrophages as well (two-way PERMANOVA test F(1,11) = 8.81; *p* = 0.004; Figure 4E).

The expression of the *Parp1* and *Parg* genes indirectly characterizes the switch to pro-inflammatory metabolism and cell energy resources since their products are involved in ADP synthesis. In addition, the *Parp1* gene product is one of the most well-known repair enzymes, and its participation in the development of a chronic inflammatory response increases the likelihood of damage to cells and DNA. The supplementation of 0.1% L-fucose into the medium for incubation with the peritoneal macrophages of the *Muc2^−/−^* and C57BL/6 mice decreased the amount of the *Parp1* gene’s mRNA in both cases (two-way PERMANOVA test F(1,12) = 6.07; *p* < 0.05; Figure 5A). At the same time, the genotype itself did not affect the *Parp1* mRNA levels (two-way PERMANOVA test F(1,12) = 0.15 ND; Figure 5A). The mRNA levels of the *Parg* gene did not differ significantly during the incubation of the peritoneal macrophages with (two-way PERMANOVA test F(1, 12) = 0.09 ND; Figure 5B) or without the addition of 0.1% L-fucose to the medium (two-way PERMANOVA test F(1, 12) = 0.09 ND; Figure 5B).

To estimate the possible pathways of peritoneal macrophage activation after the addition of L-fucose, we assessed interleukin 1 beta (Il-1β) and leukotriene C4 synthase (*Ltc4*) gene expression, as well as the gene expression of downstream targets of β-catenin such as mitochondrially encoded NADH dehydrogenase 2 (*Mt-nd2*) and mitochondrially encoded NADH dehydrogenase 6 (*Mt-nd6*). Furthermore, the expression of toll-like receptors 2 and 4 (Tlr2 and Tlr4) on the surface of the macrophages was analyzed using flow cytometry. The expression of the *Il1b* gene in the macrophages of the *Muc2^−/−^* mice was higher than that in the C57BL/6 mice’s macrophages (two-way PERMANOVA test F(1,18) = 5.5; *p* = 0.036; Figure 5C). The addition of 0.1% L-fucose to the medium for incubation with the peritoneal macrophages of the *Muc2^−/−^* and C57BL/6 mice enhanced the mRNA expression of the *Il1b* gene (two-way PERMANOVA test F(1,18) = 63.4; *p* < 0.0001; Figure 5C). The expression of the *Ltc4* gene in the peritoneal macrophages did not differ between the two genotypes and did not change after the addition of L-fucose to the medium (two-way PERMANOVA test F(1,18) = 3.6 ND; Figure 5D). Interestingly, the expression of both β-catenin downstream target genes (*Mt-nd2* and *Mt-nd6*) was higher in the peritoneal macrophages from the *Muc2^−/−^* mice (two-way PERMANOVA test F(1,18) = 5.5 and 6.8; *p* < 0.01; Figure 5E,F). The addition of L-fucose to the peritoneal macrophage culture medium downregulated the expression of both β-catenin downstream target genes in the macrophages from the *Muc2^−/−^* mice only (two-way PERMANOVA test F(1,18) = 8.9 and 8.2; *p* < 0.001; Figure 5E,F). The surface expression of Tlr2 was dependent on the addition of L-fucose (two-way PERMANOVA test F(1,18) = 9.3; *p* < 0.01; Figure 5G) but was similar in the steady state of the macrophages derived from the *Muc2^−/−^* and C57BL/6 mice (two-way PERMANOVA test F(1,18) = 3.9 ND; Figure 5G). The expression of Tlr4 on the macrophages’ surface did not differ between the two groups of mice in the steady state nor after incubation with 0.1% L-fucose (two-way PERMANOVA test F(1,18) = 0.82 and 3.2 ND; Figure 5H).

## 3. Discussion

IBD is characterized by the development of a chronic inflammatory process due to damage to the mucous layer, which protects the intestinal epithelium from contact with bacterial toxins. This interaction of bacteria and host cells leads to the activation of both innate and adaptive immune responses. In particular, macrophages play an important role in such processes [24]. Since the macrophages derived from *Muc2^−/−^* mice differ from the classic M1 and M2 types due to chronic inflammation conditions, we refer to them as M1- and M2-like.

### 3.1. The Macrophages of Muc2^−/−^ Mice Demonstrate Signs of Stress

We studied the differences between colon and peritoneal macrophages in an experimental model of IBD in *Muc2^−/−^* mice and mice without pathology. It is known that in IBD, macrophages penetrate the lamina propria of the colon mucosa, where they differentiate according to the pro-inflammatory M1 type, in contrast to tissue macrophages, which differentiate according to the M2 type under normal conditions [24]. In our work, we showed that indeed, the percentage of M1-like-type CD80^+^ peritoneal macrophages was significantly higher in the *Muc2^−/−^* mice compared to the C57BL/6 mice, and the percentage of anti-inflammatory M2-like-type CD209^+^ peritoneal macrophages was significantly lower in the experimental model of IBD.

Changes in the ultrastructures of the colon macrophages were quite expected. Pathological changes in the mitochondria were apparently associated with functional overstrain of these cells, general intoxication, and oxygen starvation, which are characteristics of an inflammatory process. At the same time, changes in the peritoneal macrophages, located distantly from the site of inflammation, were less expected. However, the ultrastructural changes in the mitochondria of the peritoneal macrophages had the same direction of changes as those in the macrophages of the intestinal mucosa, although to a significantly less extent.

### 3.2. Metabolic Reprogramming in the Muc2^−/−^ Peritoneal Macrophages

M1-type macrophages actively produce pro-inflammatory cytokines such as IL-1α, IL-1β, and IL-6, as well as ROS and reactive nitrogen species (RNS) [25]. Due to the specific conditions in the environment of inflamed tissue, such as anaerobic conditions, M1 macrophages use glycolysis and the pentose phosphate pathway as their main metabolic pathway since the Krebs cycle is disrupted at two stages [26]. The release of ROS and RNS leads to damage to the mitochondrial membranes and disruption of their functions, which is the second reason for the use of glycolysis by M1-type macrophages as their main metabolic pathway [27]. Damage to the mitochondria leads to impaired ATP synthesis, which we observed in our study according to the decrease in the amount of ATP in the peritoneal macrophages of the *Muc2^−/−^* mice compared with that in the mice from the control group. Also, disturbances in the processes of mitophagy, ROS synthesis, and mitochondrial dynamics play a role in the development of IBD [28]. The changes observed in the structure of the mitochondria of peritoneal and colon macrophages are similar to the changes observed in the epithelial cells of *Muc2^−/−^* mice [29]. This may indicate systemic disorders in the structure and function of the mitochondria as a characteristic of IBD. It is known that classic M1-type macrophages have a higher ECAR but a lower OCR [30]. The changes in respiration observed in our study are similar to the changes observed in macrophages derived from adipose tissue [31]. This may also support our hypothesis that macrophages under chronic inflammation conditions do not polarize as classic M1 and M2 types. In addition, ATP is consumed during the synthesis and excretion of IL-1β, which requires the secretion of ATP through pannexin-1 channels, with the further formation of Nlrp3 inflammasomes necessary for the maturation of IL-1β [32]. M2-type macrophages are characterized by classical oxidative phosphorylation and a complete cycle of tricarboxylic acids [22], so the amount of ATP they contain is higher, which was also observed in our study.

In addition to the regulation of gene expression and cytokine secretion, Ca^2+^ oscillations are involved in phagosome maturation, although their role in this process is not completely clear. Ca^2+^ ions enter the cell from the external environment via Gq P2Y receptors and Ca^2+^-permeable P2X channels activated by ATP. Notably, the mobilization of Ca^2+^ stored in the endoplasmic reticulum is not sufficient for caspase-1 activation and IL-1β secretion. When cells are stimulated with ATP in vitro in Ca^2+^-depleted medium, the secretion of IL-1β practically stops. These observations indicate that caspase-1 activation requires a stable and more prominent increase in intracellular Ca^2+^, which may be mediated by the activation of P2X7 channels but not of P2Y receptors [33]. In M1-type macrophages, the amount of P2X4 and P2X7 is reduced [34]. Since in our case we studied the amount of intracellular Ca^2+^, its significantly lower amount in the *Muc2^−/−^* mice may indicate a disruption in the mechanisms of Ca^2+^ entry from the external environment into the cell, which may be associated with a lack of ATP.

### 3.3. Activation of Signaling Pathways in Peritoneal Macrophages in Muc2^−/−^ Mice

TLR2 activation downstream of the TLR pathway in the macrophages enhances ROS production in the mitochondria and increases antimicrobial activity [35]. In our case, the increased percentage of TLR2-positive cells after activation with fucose might have been associated with the presence of bacterial L-fucose. This signaling may be mediated by TLR4 and CD14 and requires further study [36]. IL-1 takes part in the classic M1-type polarization via the NF-κB pathway. As expected, the IL-1 levels are increased in *Muc2^−/−^*-derived peritoneal macrophages because of chronic inflammation conditions. After L-fucose treatment, we observed an increased level of IL-1 for both genotypes. It was shown that fucose-containing mushroom extract had the same effect on the IL-1 levels in mouse spleen cells [37]. L-fucose treatment led to an increase in the level of TLR2 expression only in the *Muc2^−/−^* mice-derived peritoneal macrophages. At the same time, the expression of IL-1 was found to be increased in the macrophages of both the *Muc2*-deficient and control mice. This might be a sign of NF-κB pathway activation, but this requires further research using specific blockers of key factors in this pathway.

CD38 cleaves NAD^+^ and NADP^+^ to form the Ca^2+^-mobilizing compounds adenosine diphosphate ribose (ADPR), cyclic ADPR (cADPR), and nicotinic acid adenine dinucleotide phosphate (NAADP). In the context of the immune system, CD38 is known to induce the production of pro-inflammatory and regulatory cytokines in monocytes and dendritic cells [38]. A lack of CD38 leads to the impaired synthesis and function of TLR2 and, as a consequence, impaired recognition of pathogen-associated patterns [39]. Mice with genetic CD38 knockout do not develop colitis when exposed to LPS [40]. Our work showed a significantly higher expression of the intracellular marker CD38 in the peritoneal macrophages of the *Muc2^−/−^* mice compared to that in the peritoneal macrophages of the mice in the control group, which is consistent with data on studies of colon tissues in patients [41]. The depletion of NAD^+^ after ischemic stroke can lead to cell death and is associated with the excessive activation of poly-ADP-ribose polymerase (PARP1), as well as an increase in the CD38 enzyme, which consumes NAD^+^ [42].

An altered mitochondrial membrane potential (MMP) and the formation of ROS are often associated with macrophage pyroptosis [43]. Moreover, mitochondrial stress can induce mitophagy to eliminate dysfunctional mitochondria, characterized by a low membrane potential and high ROS levels. If the damage can not be repaired, the stressed macrophages undergo apoptosis [44]. After LPS stimulation, macrophages redirect their mitochondria from ATP production to succinate-dependent ROS formation, while glycolysis plays the role of ATP formation, allowing the mitochondria to maintain a high membrane potential [45]. Thus, a high membrane potential should also be maintained in M1-activated macrophages, despite changes in their metabolism. However, we observed a decrease in the number of peritoneal macrophages containing mitochondria with a high membrane potential in the *Muc2^−/−^* mice compared to the WT mice, despite their M1-type polarization. Due to the fact that altered mitochondrial function in the intestinal epithelial cells of *Muc2^−/−^* mice has previously been demonstrated [15], it can be assumed that these alterations are more systemic than them solely being associated with macrophage polarization.

Mitochondrial dysfunction exacerbates inflammatory signaling in several ways. Damaged mitochondrial DNA is a damage-associated molecular template that activates the NLRP3 inflammasome; priming and activation of the NLRP3 inflammasome and hence the release of IL-1β and IL-18 from gasdermin D pores cause increased inflammatory responses. The increase in ROS caused by mitochondrial damage also activates inflammatory pathways, while blocking complex enzymes is sufficient to increase inflammatory signaling. The disruption of mitophagy promotes inflammation because failure to renew the mitochondria in a timely manner increases the levels of ROS and damaged mtDNA, with the latter likely stimulating the cGAS-STING pathway to enhance interferon signaling. In addition to dysfunctional mitochondria, which increase inflammation, inflammation also reduces mitochondrial function and ATP production, which accelerates degeneration and leads to the development of a chronic inflammatory process [46].

MUC2 treatment mitigated the pro-inflammatory response in antigen-presenting cells via β-catenin [14]. Mitochondrially encoded NADH dehydrogenase 2 and 6 (*Mt-Nd2* and *Mt-Nd6*) are included in the normal metabolic process of the mitochondria and are targets of the β-catenin pathway [15]. However, an increase in their expression levels might be significant to inflammatory processes. *Mt-Nd2* is crucial for controlling the production of ROS, and single-nucleotide polymorphisms in *Mt-Nd2* suppress ROS production and type 1 diabetes development [47]. M1-type macrophages are well known for their ROS production, and macrophages of the M1-like type have shown signs of oxidative stress in their changes in mitochondrial structure and respiration. *Mt-Nd6* is associated with the pro-inflammatory response and immune regulation and was upregulated in T- and B-cells in chronic antibody-mediated rejection after renal transplantation [48]. Also, higher expression levels of *Mt-Nd6* are associated with a higher OCR in cardiac mesenchymal stem cells [49]. That *Mt-Nd2* and *Mt-Nd6* levels decreased in the *Muc2^−/−^*-derived peritoneal macrophages after the addition of L-fucose may have been an indication of the downregulation of the β-catenin pathway, but this requires further investigation.

Mucin-2 is the main mucus protein that protects the colon’s epithelial cells from damaging factors that lead to the activation of the immune system. In addition, mucin-2 is glycosylated by fucose in the terminal positions. So, protection by mucus from the external environment can be supplemented by the effect of fucose on epithelial cells.

In our work, it was shown that direct exposure to 0.1% L-fucose during co-incubation with the peritoneal macrophages significantly reduced the amount of Ca^2+^ for both genotypes. Anti-inflammatory macrophages of the M2 type do not receive a signal to open their Ca^2+^ channels in dependence on extracellular ATP [50], so we suggest that the decrease in the level of intracellular Ca^2+^ in the peritoneal macrophages may have been associated with their differentiation into the M2 type under the influence of L-fucose.

The high activity of the transcription processes of pro-inflammatory cytokines in *Muc2^−/−^* mice leads to DNA damage, including the occurrence of single- and double-strand breaks [51]. During chronic inflammatory processes, iNOS induction leads to the synthesis of ROS and RNS, which also leads to DNA damage [52]. Poly-ATP-ribosylation of nucleotides regulates DNA repair processes [53] since the correct response to DNA damage requires both timely and ordered production of poly-ATP-ribose by PARP-1 and its degradation by PARG [54]. A decrease in the mRNA levels of the *parp1* gene in the peritoneal macrophages after their co-incubation with L-fucose may indicate a decrease in the amount of DNA damage, including due to the absence of the ROS and RNS characteristic of M1 macrophages.

## 4. Materials and Methods

### 4.1. Animals and Housing Conditions

This study was conducted in the Scientific Research Institute of Neurosciences and Medicine (SRINM). All of the procedures were carried out in accordance with the European Directive 86/609/EEC (Council of the European Communities, 1986) and the European Convention for the Protection of Vertebrate Animals used for Scientific Purposes. All manipulations with animals were approved by the SRINM ethics committee. Animals free of specific pathogens (FELASA 2014 annual list) were used in this study, with exception of the presence of *Pasteurella pneumotropica*.

In the experiments, 3-month-old, approximately 20 to 25 g weight female mice of the inbred line C57BL/6JNsknm (referred to as C57BL/6 elsewhere) and *Muc2^−/−^* knockout mice were used. We used female mice because the male mice obtained from breeding were used in other experiments. The C57BL/6 and *Muc2^−/−^* mice were obtained from the Center for Genetic Resources of the Institute of Cytology and Genetics of the Siberian Branch of the Russian Academy of Sciences. After crossing heterozygous *Muc2^−/−^* mice, homozygous *Muc2^−/−^* mice were obtained. The mice were kept in same-sex groups of 3–6 animals in open cages under a light regime of 12C:12T, at a temperature of 20–22 °C, and at a humidity of 36%. Dust-free birch shavings were used as their bedding. The animals received food and clean drinking water given *ad libitum*.

### 4.2. Sample Collection

For peritoneal macrophage isolation, the animals were euthanized by decapitation at the age of 12 weeks when they had histological signs of IBD, as shown in previous works [11]. Next, skin from the abdominal region was removed, and 5 mL of sterile, ice-cold PBS was injected into the peritoneal cavity. After gently massaging the peritoneum, a suspension of peritoneal cells was collected using a syringe. Then, the cell suspension was centrifuged for 5 min at 1500 rpm and at RT. After the removal of the supernatant, the cell pellets were resuspended in DMEM (Gibco, Waltham, MA, USA) containing 10% FBS (Gibco, Waltham, MA, USA) and seeded onto a 24-well culture plate (Corning, Corning, NY, USA) at a density of 1 × 10^6^ cells per well. As macrophages tend to attach to plastic, unlike other cells, the suspension was left to adhere for 1 h in a CO_2_ incubator at 37 °C. To remove non-adherent cells, the wells were washed twice with PBS (Gibco, USA). Adherent macrophages were used for further experiments. To analyze the influence of fucose on cellular metabolism and phagocytosis, the peritoneal macrophages collected from each mouse were divided into two wells and incubated with or without 0.1% L-fucose for 12 h. Then, the cells were washed with supplement-free medium and used for the downstream experiments.

### 4.3. Analysis of Gene Expression in the Peritoneal Macrophages

The gene expression of several inflammation-associated genes of the peritoneal macrophages was analyzed using real-time qPCR. The peritoneal macrophage samples were homogenized in 500 μL of TriReagent (Sigma-Aldrich, Burlington, MA, USA). Then, 100 μL of chloroform was added, and the solution was mixed and centrifuged for 10 min at 12,000× *g* and 4 °C. The upper aqueous phase was transferred into a new tube, and the RNA was precipitated with an equal amount of isopropanol for 10 min on ice and then centrifuged at 4 °C for 15 min at 12,000× *g*. Next, the pellet was washed once with 1 mL of 75% ethanol. Then, the pellet was dried and dissolved in deionized water. In order to remove the genomic DNA; the samples were then treated with DNase I (Thermo Scientific, Waltham, MA, USA) according to the manufacturer’s protocol. Next, the RNA was again precipitated with 96% ethanol at 20 °C for 2 h and then centrifuged at 4 °C for 15 min at 12,000 rpm. After subsequent washing with 70% ethanol, the samples were dried and dissolved in deionized water. The RNA concentration was determined using a NanoDrop 2000 (Thermo Scientific, Waltham, MA, USA). A total of 3–7 μg of RNA was used in a reverse transcription reaction, which was performed with the MulV reverse transcriptase kit (SibEnzyme, Novosibirsk, Russia) according to the manufacturer’s recommendations. A mixture of RNA, a set of primers (oligodT and random primers, 600 ng each), and dNTPs (1 mM each) was heated to 65 °C for 5 min and then cooled down on ice for 5 min. Next, reverse transcriptase (100 UA) and the reaction buffer were added to the reaction mixture. The mixture was incubated at 37 °C for an hour and then heated to 70 °C for 5 min for enzyme inactivation. The obtained cDNA samples were stored at –20 °C before their use in further experiments. The amount of the genes’ mRNA was measured using real-time qPCR. Primer sets were designed in our laboratory (Table 1). The reaction mixture contained BioMaster HS-qPCR SYBR Blue 1Х (BioLabMix, Novosibirsk, Russia), specific primers (0.3 μM each), and 40–200 ng of cDNA per the reaction volume of 20 μL. The real-time qPCR was performed on a CFX96 Touch™ Deep Well Real-Time PCR Detection System (Bio-Rad Laboratories, Hercules, CA, USA). The thermocycling conditions were as follows: 95 °C for 5 min and 40 cycles of [95 °C for 15 s, 62 °C for 25 s, and 72 °C for 25 s]. The length of the resultant DNA product was verified using agarose gel electrophoresis. Each sample was run in triplicate, and the average value was presented as a result. The differences between triplicates were less than 5%. The amount of specific gene mRNA was normalized to the amount of beta-tubulin (Tubb) mRNA using the formula 2^(Ct of target gene − Ct of the Tubb gene)^, where Ct is the cycle threshold value for each sample.

### 4.4. Flow Cytometry Analysis

Adherent macrophages were collected, washed twice in wash buffer (2% FBS in PBS), and stained with FITC-F4/80 (Rat IgG2b, k, clone EMR1), PE-CD209a (Mouse IgG2c, clone MMD3), and FITC-anti-CD80 (Rat IgG2a, clone GL-1) (all BioLegend, San Diego, CA, USA) for 60 min at 4 °C in the dark. After washing them, the cells were resuspended in staining buffer (1% BSA, 0.1% sodium azide in PBS) and analyzed using a BD FACSCanto II Flow Cytometer (BD Biosciences, Franklin Lakes, NJ, USA). For the analysis, we first gated for a single cell population (singlets), and then the F4/80 marker was used to identify the macrophages. This cell population was further used to identify the counts of the M1-like (positive for CD80) and M2-like (positive for CD209) macrophage subtypes. For the analysis of CD44^+^CD11b^+^CD38^+^ cells, the cells were fixed with True-Nuclear™ Fix reagent (BioLegend, San Diego, CA, USA) for 50 min in the dark at room temperature. Then, the cells were permeabilized with True-Nuclear™ Perm Buffer (BioLegend, San Diego, CA, USA). After centrifugation, the cells were resuspended in 100 μL of permeabilization buffer and incubated with Pacific Blue–anti-CD45 (BioLegend, San Diego, CA, USA, Rat IgG2b, k, clone S18009F), PE-anti-CD11b (BioLegend, USA, Rat IgG2b, k, clone M1/70), and FITC-anti-CD38 (Invitrogen, Waltham, MA, USA, clone 90) for 60 min at 4 °C in the dark. After washing them in the permeabilization buffer, the cells were resuspended in the staining buffer and analyzed using a BD FACSCanto II flow cytometer. For the analysis, we first identified the single cell population, and then the CD45^+^ cell population was determined as myeloid cells, among which the macrophages were identified as CD11b^+^ cells. This cell population was further used to identify CD38^+^ macrophages.

The isotype controls for all of the antibodies used are presented in Appendix A.

### 4.5. ΔѰm Analysis

The JC-1 Mitochondrial Membrane Potential Assay Kit (Servicebio, Wuhan Optics Valley Biolake, Wuhan, China) was used to detect cells containing mitochondria with a high membrane potential. A total of 500 μL of a mixture consisting of 4 μL of JC-1 dye (500×) and 900 μL of JC-1 diluent was added to the cells in the medium and left for 30 min at 37 °C in a CO_2_ incubator. Then, the cells were washed twice with JC-1 buffer and centrifuged at 1500× *g* for 6 min, the supernatant was removed, and the cells were resuspended in 200 μL of staining buffer (1% bovine serum albumin (BSA) and 0.1% sodium azide in PBS). The samples were analyzed using a BD FACSCanto II flow cytometer (BD Biosciences, Franklin Lakes, NJ, USA). Peritoneal macrophages with fluorescence in the PE channel were considered to be macrophages containing mitochondria with a high membrane potential, while cells with fluorescence in the FITC channel were considered to be macrophages containing mitochondria with a low membrane potential.

### 4.6. Analysis of Intracellular ATP Concentration

A total of 40 µL of luciferase (Hygiena, Camarillo, CA, USA) was added to 5 µL of the peritoneal macrophage cell suspension in a 96-well black opaque plate. Then, the luminescence was measured using the TriStar LB 941 (Berthold Technologies, Bad Wildbad, Germany). Sodium ATP solutions of a known concentration were used for the generation of calibration curves. After the addition of the luciferase agent, analysis of the standard samples was carried out in parallel with the experimental samples. The ATP concentration in the samples was normalized to the protein concentration, which was measured using the Bradford assay [55] and presented as ng of ATP per μg of protein.

### 4.7. Analysis of Intracellular Calcium Concentration

A total of 100 µL of Reagent 1 and 100 µL of Reagent 2 (Olvex diagnostic, Saint Petersburg, Russia) was added to 5 µL of the peritoneal macrophage cell suspension in 96-well transparent plates. The absorbance was measured using a Microplate Absorbance Reader (Bio-Rad Laboratories, Hercules, CA, USA) at 570 nm. The concentration of Ca^2+^ in the samples was calculated as C = Ab(sample)/Ab(calibration sample) × 2.5 and normalized to the protein concentration, which was measured using the Bradford assay [55] and presented as mmol of Ca^2+^ per mg of protein.

### 4.8. TEM

Samples from the descending colon (N = 3 for C57BL/6 and *Muc2^−/−^* mice) were fixed in 2.5% glutaraldehyde solution in 0.1 M sodium cacodylate buffer (pH 7.4) for 1 h at room temperature, washed thrice in the same buffer, and postfixed in 1% OsO4 with 0.8% potassium ferrocyanide in cacodylate buffer for 1 h. The samples were rinsed thrice in distilled water, incubated in 1% aqueous solution of uranyl acetate, dehydrated in an ethanol series and acetone, and embedded into epoxy resin (Epon 812). The peritoneal macrophages were fixed and processed in a similar manner to that adapted for the cell suspensions and were also enclosed in Epon 812 [56]. Semi-thin cross-sections were sliced, stained with methylene blue, and analyzed using an Axioscope-4 microscope (Carl Zeiss, Oberkochen, Germany). Ultra-thin sections (60 nm) for TEM were cut with a diamond knife on a Leica EM UC7 ultramicrotome (Leica, Wetzlar, Germany) and then analyzed using a JEM1400 transmission electron microscope (JEOL, Akishima, Japan) with an operating voltage of 80 kV. We used the Specimen Quick Change Holder EM-11210SQCH (JEOL, Akishima, Japan). TEM was performed at the Interinstitutional Shared Center for Microscopic Analysis of Biological Objects (ICG SB RAS, Novosibirsk). In the sections, the mitochondria in the macrophages of the colonic lamina propria were measured. For the statistical analysis, the morphological structures were measured using iTEM 5.1 software (OlympusSIS, Hamburg, Germany) in randomly chosen sections (80 to 100 independent measurements per genotype) [18].

In the mitochondria, the length, width, perimeter, surface area, total length of cristae, number of cristae, and area of “empty” spaces were measured. Structural alterations in the mitochondria were categorized as follows:Normal structures;“Empty” spaces of the light matrix without cristae;“Hernias”: protrusions or invaginations of the mitochondria’s outer membrane;Ruptures of the outer membrane of the mitochondria.

The roundness coefficient for the mitochondria was calculated as the ratio of the short side to the long side. The surface density of the cristae was measured as the ratio of the number of cristae to the area of the mitochondria (cristae per µm^2^). In addition, the percentage of the area of “empty” space to the total area of the mitochondria was calculated.

### 4.9. Mitochondrial Respiration and Anaerobic Glycolysis

The oxygen consumption rate (OCR) and extracellular acidification rate (ECAR) were evaluated using the Seahorse XF Cell Mito Stress Test (Agilent, Santa Clara, CA, USA). C57BL/6 (*n* = 5) and *Muc2^−/−^* (n = 6) female mice were used for the isolation of the peritoneal macrophages. After seeding them onto a plate, the samples were incubated for 12 h at 5% CO_2_ and 37 °C. Next, the cells were transferred onto a XFe24 Bioflux plate (Seahorse Bioscience, North Billerica, MA, USA) at a concentration of 5 × 10^5^ cells/well. Mitochondrial function was assessed as the OCR after the addition of 0.5 μM of oligomycin (an ATP synthase inhibitor), 1 μM of FCCP (an electron transport chain accelerator), and 1 μM antimycin A (a complex III inhibitor) with 1 μM of rotenone (a complex I inhibitor), according to the manufacturer’s instructions. Seahorse XFe Wave Software 2.6 (Seahorse Bioscience, North Billerica, MA, USA) was used for the data analysis.

### 4.10. Fluorescent Microscopy

For the microscopic analysis of the peritoneal macrophage samples, direct immunostaining was used. The macrophages were stained with Alexa Fluor 568 Phalloidin (#A12380, ThermoFisher Scientific, USA) and DAPI (#D1306, ThermoFisher Scientific, USA). After PBS washing, 5 µL of the macrophage cell suspension was transferred onto a glass slide and covered with a cover glass. The samples were analyzed on a LSM 710 confocal microscope (Carl Zeiss, Oberkochen, Germany).

The gut samples were fixed in 3.7% formaldehyde and then kept in 15% sucrose for 24 h and in 30% sucrose for another 24 h. Sucrose-protected tissue was embedded into Tissue-Tek^®^ O.C.T. Compound 4583 (Sakura, Torrance, CA, USA) and quick-frozen in liquid nitrogen. Sections of a 40 μm thickness were prepared using a CM1850 UV cryostat (Leica, Wetzlar, Germany). Non-specific binding was blocked through incubation in 0.5% BSA and 0.3% Triton X-100 in PBS for 1 h at 4°С. Next, the samples were incubated overnight at 4 °C with the desired primary antibodies conjugated to fluorescent dyes. In this study, the following antibodies were used: FITC anti-mouse F4/80 (#123108, BioLegend, USA), FITC anti-mouse CD80 (#104708, BioLegend, USA), and PE anti-mouse CD209a (#833004, BioLegend, USA). After incubation, the samples were transferred onto glass slides in 50% glycerol in PBS containing DAPI/Antifade Solution and covered with a cover glass. Imaging was performed using an LSM 510 Meta (Zeiss, Germany) and using ZEN 2007 2.6 lite software.

### 4.11. Multiplex Cytokine Analysis

To determine the levels of 32 cytokines/chemokines (G-CSF, GM-CSF, IFN-γ, IL-1α, IL-1β, IL-2, IL-3, IL-4, IL-5, IL-6, IL-7, IL-9, IL-10, IL-12 (p40), IL-12 (p70), IL-13, IL-15, IL-17, IP-10, KC, LIF, LIX, MCP-1, M-CSF, MIG, MIP-1, MIP-2, RANTES, TNF-α, VEGF, and Eotaxin/CCL11), the supernatant was collected from the plates after the incubation of the peritoneal macrophages overnight at 37 °C in a CO_2_ incubator. Then, it was quickly frozen in liquid nitrogen and stored at −70 °C until the analysis. A total of 100 μL of the supernatant was taken for analysis using the commercially available MILLIPLEX^®^ MAP Mouse Kit (Merck, Darmstadt, Germany) and the Cytokine/Chemokine Magnetic Bead Panel (Merck, Darmstadt, Germany). A total of 32 analytes were determined according to the manufacturer’s instructions. The fluorescence analysis was performed on a Luminex 200 instrument (Merck, Darmstadt, Germany). The amount of cytokines was normalized to the amount of protein determined using the Bradford method [56] and expressed as pg/mg protein.

### 4.12. Oxygen Consumption Rate

The oxygen consumption rate (OCR) was measured using the Seahorse XF Cell Mito Stress Test (Agilent, Santa Clara, CA, USA). The analysis was performed according to the manufacturer’s instructions. First, the peritoneal macrophages were incubated for 12 h in a CO_2_ incubator at 37 °C. Then, the cells were then seeded onto an XFe24 Bioflux plate (Seahorse Bioscience, North Billerica, MA, USA) at a concentration of 5 × 10^5^ cells/well. The functional activity of the mitochondria was measured as the OCR after the addition of 0.5 μM of oligomycin (an ATP synthase inhibitor), 1 μM of FCCP (carbonyl cyanide-p-trifluoromethoxyphenylhydrazone) (an electron transfer chain accelerator), and 1 μM of antimycin A (a complex III inhibitor), plus 1 μM of rotenone (a complex I inhibitor), according to the manufacturer’s instructions. For the data analysis, Seahorse XFe Wave (Seahorse Bioscience, USA) was used.

### 4.13. Statistical Analysis

Statistical analysis of the data obtained was carried out on IBM SPSS Statistics software 26.0.0.1. Comparison between two groups of data with normal distribution was performed using Student’s *t*-test, and for data with a non-normal distribution, the Mann–Whitney test was used. Comparison between multiple groups of data with a non-normal distribution was performed using a PERMANOVA. All of the data are presented as means ± standard deviation.

## 5. Conclusions

*Muc2^−/−^* mice were shown to exhibit lower levels of intracellular ATP and Ca^2+^ in the peritoneal macrophages, a higher percentage of star-shaped macrophages, and an increased oxygen consumption rate (OCR), combined with a higher proportion of mitochondria exhibiting an impaired structure, compared to C57BL/6 mice. The peritoneal macrophages from the *Muc2^−/−^* mice expressed pro-inflammatory cytokines including IL-1α, IL-6, MIP-1a, MIP-1b, and KC, indicating polarization towards an M1-like phenotype. A reduction in cell cytoplasm size and an increased proportion of mitochondria with signs of condensation further supported the polarization into the M1-like phenotype and cellular stress. The addition of 0.1% L-fucose to the peritoneal macrophages in vitro eliminated the differences in the cytokine expression levels between the two genotypes and reduced the intracellular Ca^2+^ levels in the peritoneal macrophages of both genotypes, which may indicate a shift towards M2-like polarization. Additionally, the L-fucose treatment reduced the expression of the *Parp1* gene and genes associated with the β-catenin pathway in the peritoneal macrophages of both genotypes. This study provides insight into the potential mechanisms of action of L-fucose in activated macrophages from *Muc2^−/−^* mice. However, to elucidate the pathways involved in macrophage activation and the broader effects of L-fucose further, similar studies should be conducted using macrophages obtained from other experimental models of IBD.

## Figures and Tables

**Figure 1 ijms-26-00013-f001:**
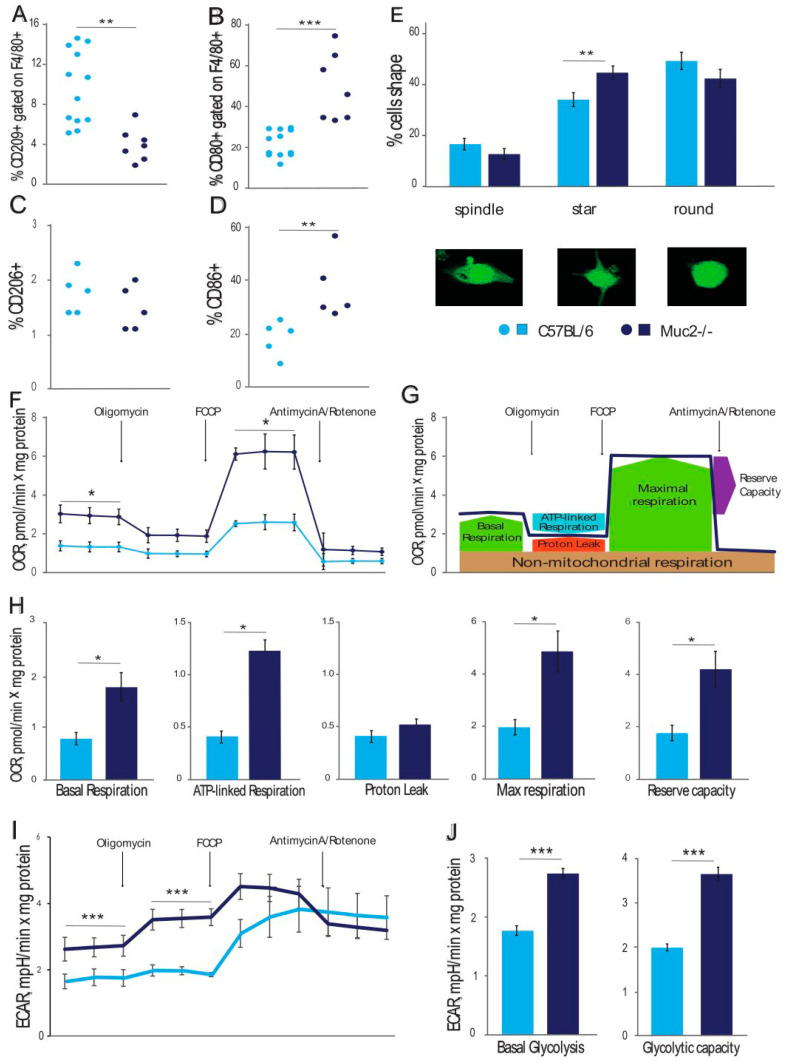
Comparison of marker expression, shapes, OCR, and ECAR of peritoneal macrophages derived from C57BL/6 and *Muc2^−/−^* mice. (**A**). Percentage of CD209^+^ macrophages (M2-like type) from two mouse strains. (**B**). Percentage of CD80^+^ macrophages (M1-like type) from two mouse strains. (**C**). Percentage of CD206^+^ macrophages (M2-like type) from two mouse strains. (**D**). Percentage of CD86^+^ macrophages (M1-like type) from two mouse strains. (**E**). Percentage of different shapes of macrophages from two mouse strains. (**F**). The OCRs were measured following treatment with oligomycin, FCCP, and the antimycin A/rotenone complex; (**G**). Seahorse assay scheme showing the types of respiration. (**H**). Means of different types of respiration: basal, ATP-linked, proton leak, maximal respiration, and reserve capacity; (**I**). The ECAR was measured following the treatment with oligomycin, FCCP, and the antimycin A/rotenone complex. (**J**). Means of basal glycolysis and glycolytic capacity. “C57BL/6” vs. “*Muc2^−/−^*”: * *p* < 0.05, ** *p* < 0.01, and *** *p* < 0.001 according to the PERMANOVA test.

**Figure 2 ijms-26-00013-f002:**
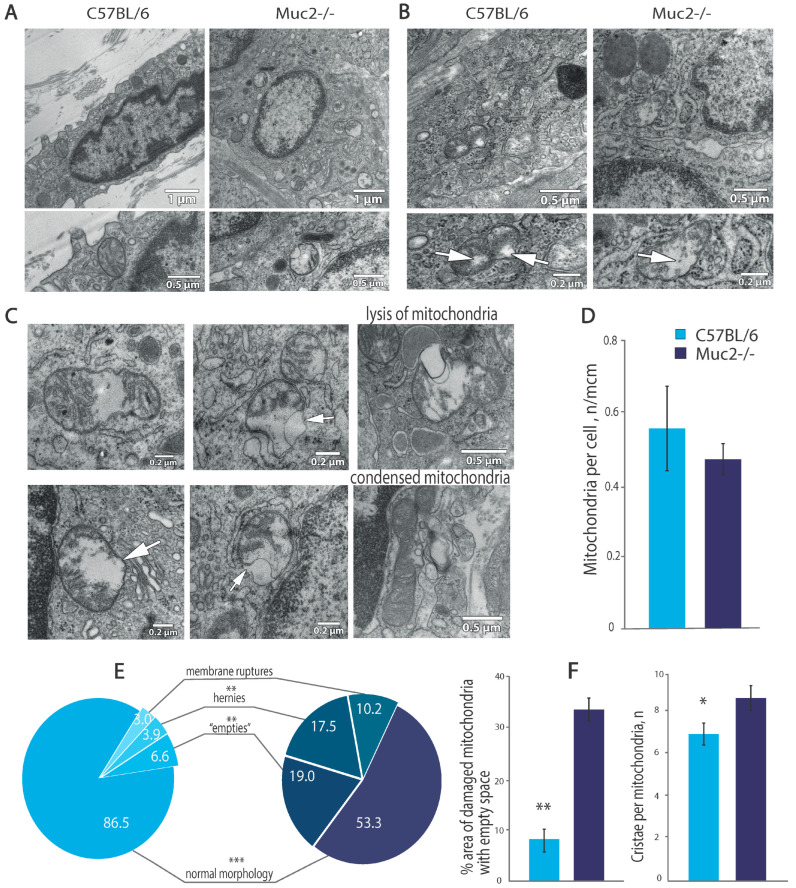
Ultrastructures of macrophages from the lamina propria of colons derived from C57BL/6 and *Muc2^−/−^* mice. (**A**). Morphology of the macrophages from the two mouse strains. *Muc2^−/−^* mice macrophages have a lot of altered mitochondria. (**B**). Mitochondria with “empties” (white arrows show the “empties”) (**C**). Different structural defects and functional types of ultrastructures in the mitochondria in the lamina propria of the colons of the *Muc2^−/−^* mice, white arrows show the defects (**D**). The number of mitochondria in 1 μm^2^ of the cytoplasm. (**E**). Percentages of mitochondria with normal and altered ultrastructures in *Muc2^−/−^* (dark blue) and C57BL/6 (light blue) mice (**F**). Percentages of mitochondria with empty spaces and the number of cristae per mitochondria; “C57BL/6” vs. “*Muc2^−/−^*”; * *p* < 0.05, ** *p* < 0.01 and *** *p* < 0.001; *t*-test.

**Figure 3 ijms-26-00013-f003:**
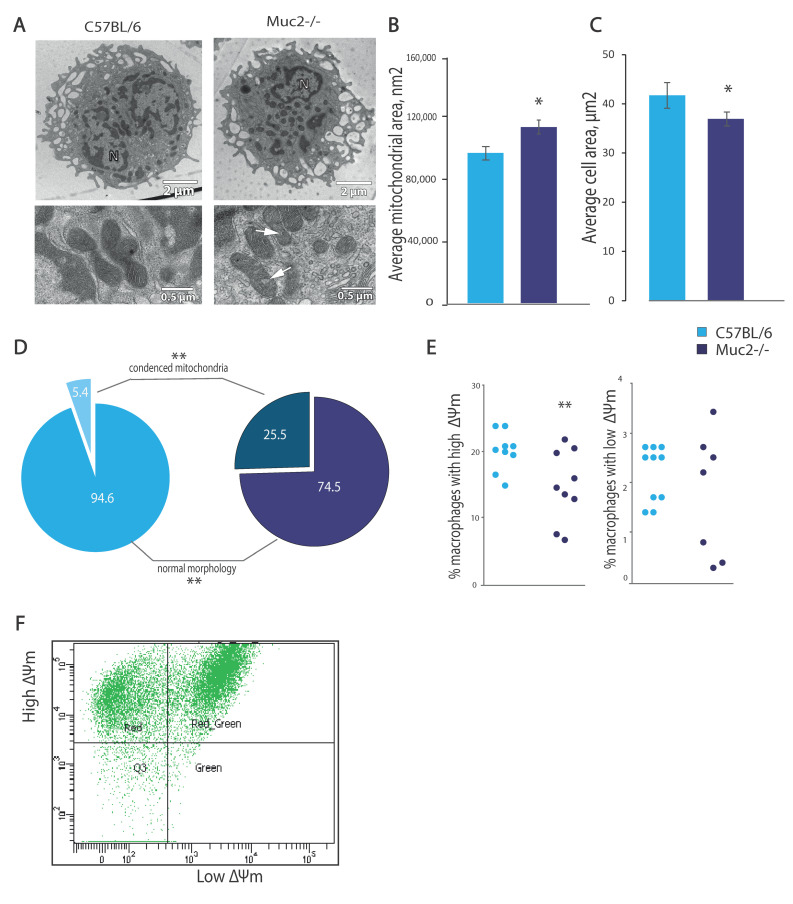
Ultrastructure and mitochondrial membrane potential of peritoneal macrophages derived from C57BL/6 and *Muc2^−/−^* mice. (**A**). Morphology of the peritoneal macrophages from the two mouse strains. white arrows show dilated cristae in condensed mitochondria. (**B**). Morphology of the mitochondria in the peritoneal macrophages from the two mouse strains. (**C**). Size of peritoneal macrophages and the mitochondria in them in the two mouse strains. (**D**). Percentages of mitochondria with normal and altered ultrastructures in the *Muc2^−/−^* (dark blue) and C57BL/6 (light blue) mice. (**E**). A FACS graph with the percentage of peritoneal macrophages with a different mitochondrial membrane potential (ΔѰm) analyzed using JC-1 staining. (**F**). Percentages of peritoneal macrophages with a high and low mitochondrial membrane potential (ΔѰm). “C57BL/6” vs. “*Muc2^−/−^*”; * *p* < 0.05, ** *p* < 0.01; *t*-test.

**Figure 4 ijms-26-00013-f004:**
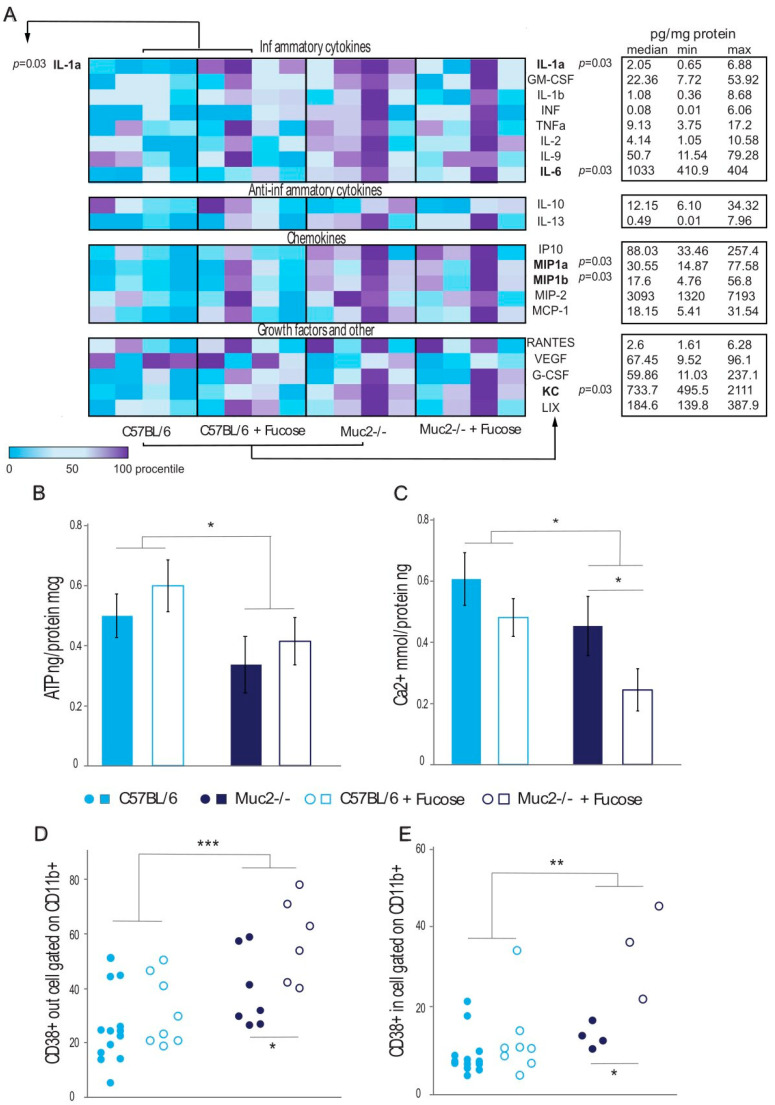
Addition of fucose to the cell culture medium affected the production of cytokines, ATP and Ca^2+^ levels, and surface and intracellular expression of CD38 of peritoneal macrophages derived from C57BL/6 and *Muc2^−/−^* mice. (**A**). Inflammatory cytokine levels in cell culture medium of peritoneal macrophages from the two mouse strains with and without the addition of 0.1% L-fucose; the median, min, and max for each cytokine is presented in pg/mg protein. (**B**). ATP levels in the peritoneal macrophages of two mouse strains incubated with and without 0.1% L-fucose. (**C**). Ca^2+^ levels in the peritoneal macrophages of the two mouse strains incubated with and without 0.1% L-fucose. (**D**). Percentage of peritoneal macrophages with surface CD38 expression from the two mouse strains incubated with and without 0.1% L-fucose. (**E**). Percentage of peritoneal macrophages with intracellular CD38 expression from the two mouse strains incubated with and without 0.1% L-fucose. “C57BL/6” vs. “*Muc2^−/−^*“ and “with 0.1% L-fucose” vs. “without 0.1% L-fucose”: * *p* < 0.05, ** *p* < 0.01, and *** *p* < 0.001. Two-way PERMANOVA test.

**Figure 5 ijms-26-00013-f005:**
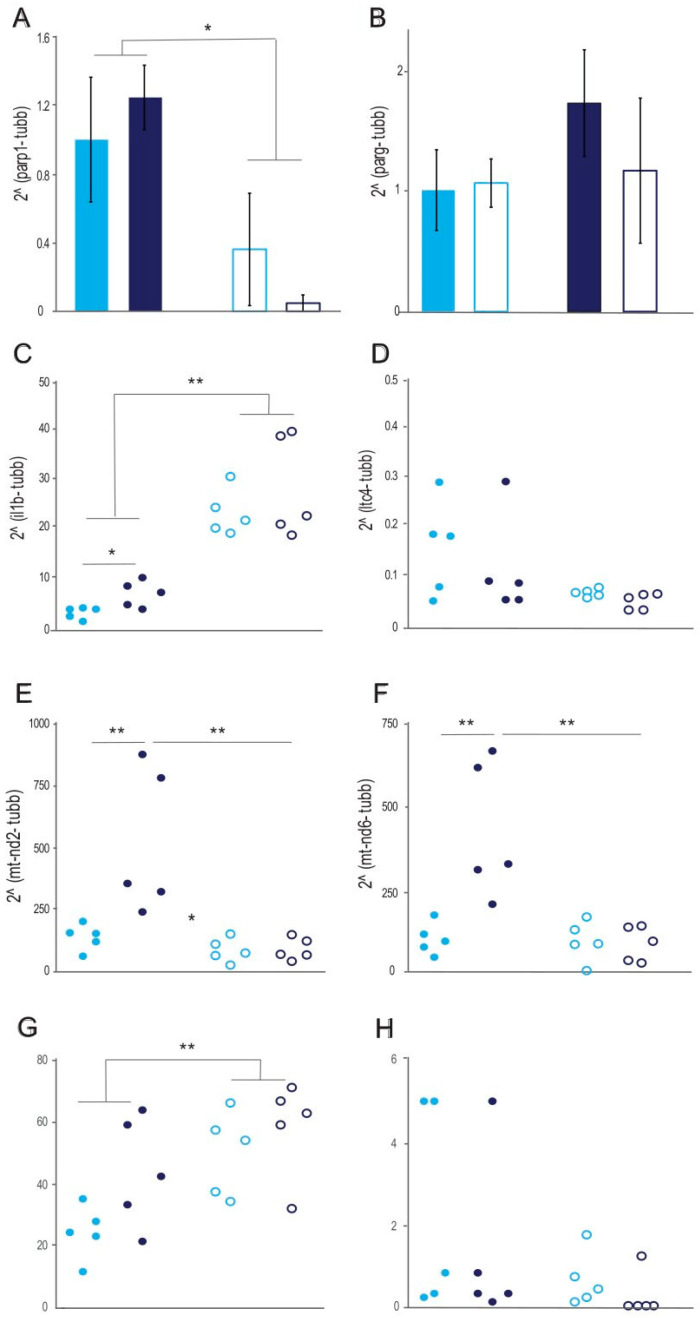
Effect of fucose on the expression of the *Parp1*, *Parg*, *Il1b*, *Ltc4*, *Mt-nd2*, and *Mt-nd6* genes and on Tlr2 and Tlr4 expression on the surface of the peritoneal macrophages. (**A**). Expression of the *Parp1* gene in peritoneal macrophages from the two mouse strains incubated with and without 0.1% L-fucose. (**B**). Expression of the *Parg* gene in peritoneal macrophages from the two mouse strains incubated with and without 0.1% L-fucose. (**C**). Expression of the *Il1b* gene in peritoneal macrophages from the two mouse strains incubated with and without 0.1% L-fucose. (**D**). Expression of the *Ltc4* gene in peritoneal macrophages from the two mouse strains incubated with and without 0.1% L-fucose. (**E**). Expression of the *Mt-nd2* gene in peritoneal macrophages from the two mouse strains incubated with and without 0.1% L-fucose. (**F**). Expression of the *Mt-nd6* gene in peritoneal macrophages from the two mouse strains incubated with and without 0.1% L-fucose. (**G**). Percentage of Tlr2-positive peritoneal macrophages from the two mouse strains incubated with and without 0.1% L-fucose. (**H**). Percentage of Tlr4-positive peritoneal macrophages from the two mouse strains incubated with and without 0.1% L-fucose. “C57BL/6” vs. “*Muc2^−/−^*” and “with 0.1% L-fucose” vs. “without 0.1% L-fucose”: * *p* < 0.05 and ** *p* < 0.01. Two-way PERMANOVA test.

**Table 1 ijms-26-00013-t001:** Primers.

Target	Primer Name	Primer Sequence 5′→3′
Mouse *Parp1*	Parp1_F	GCAGCGAGAGTATTCCCAAG
	Parp1_R	CCGTCTTCTTGACCTTCTGC
Mouse *Parg*	Parg_F	CTGTTCACTGAGGTGCTGGA
	Parg_R	TCTCAGGCACAAACTGATCG
Mouse *Mt-Nd2*	ND2_F ND2_R	AGGGGCATGAGGAGGACTTA TTGAGTAGAGTGAGGGATGGGT
Mouse *Mt-Nd6*	ND6_F ND6_R	CAACGCCTGAGCCCTACTAA AGGACTGGAATGCTGGTTGG
Mouse *Ltc4s*	Ltc4_F Ltc4_R	TCTTCCGAGCCCAGGTAAACT GACTAGCAAGCCCAGTGCAG

## Data Availability

The data presented in this study are available on request from the corresponding author.

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
