# Peer review of "Changes in the Phenotype and Metabolism of Peritoneal Macrophages in Mucin-2 Knockout Mice and Partial Restoration of Their Functions In Vitro After L-Fucose Treatment"

_ijms, 2024, doi:10.3390/ijms26010013_

Round 1

Reviewer 1 Report

Comments and Suggestions for Authors

There are the following items that should be revised.

1.       The abstract must contain an introduction, objectives, methodology, main results/findings, and conclusions (even for bibliographic works), and the results should be supported by specific data.

2.       The Introduction is a little long. Please make it simpler and more logical.

3.       For all figures, All the numbers of the coordinate axis and axis title are not clear, please use black font.

4.       About TEM, what is the sample holder, and how about the operating voltage of TEM?

5.       Conclusion is a little bit lengthy, and lack of logic, please revise it and state the limitations of their study (if any).

Author Response

Dear reviewer!

We are grateful for your revision, which has greatly improved our work.

Comment 1: The abstract must contain an introduction, objectives, methodology, main results/findings, and conclusions (even for bibliographic works), and the results should be supported by specific data.

Response 1: Thank you for pointing this out. We agree with this comment. Therefore, we have completely revised the abstract

Comment 2: The Introduction is a little long. Please make it simpler and more logical   

Response 2: We agree with this comment, thank you. We have, accordingly, modified the Introduction

Comment 3: For all figures, All the numbers of the coordinate axis and axis title are not clear, please use black font.

Response 3: Thank you for pointing this out. Done

Comment 4: About TEM, what is the sample holder, and how about the operating voltage of TEM?

Response 4: We used the Specimen Quick Change Holder EM-11210SQCH and 80kV operating voltage

Comment 5: Conclusion is a little bit lengthy, and lack of logic, please revise it and state the limitations of their study (if any).

Response 5: Thank you for this note. We have revised the Conclusion to make it shorter and more logical

Reviewer 2 Report

Comments and Suggestions for Authors

Dear Editors,

In the manuscript "Metabolism of colon and peritoneal macrophages during intestinal inflammation of Muc2-/- mice and L-fucose effect in vitro", the authors aim to demonstrate that L-fucose can influence the inflammatory activity of peritoneal macrophages derived from Muc2-/- mice  (in vivo experimental model of IBD). Although the authors provide several experimental approaches to validate their hypotheses, the manuscript needs extensive revisions to make it suitable for publication on IJMS.

Comments:

1.      Substantial revisions of introduction should be performed to make the aims of the work and the experimental strategies more comprehensible to the readers.

2.      Although the authors have reported the data about OCR, IHC and the level of secreted cytokines analyses, the corresponding techniques are not described in the manuscript. It is necessary to integrate them in the materials and methods section.

3.      The Figure 4 pdf file is not attached to the manuscript and the captions should be included also in the supplementary figures. Furthermore, a list of acronyms should be included in the manuscript.

Comments on the Quality of English Language

     I suggest improving English to more clearly express the research results.

Author Response

Dear reviewer,

We are grateful for your interest in our work and valuable comments, which allowed us to significantly improve our manuscript.

Comment 1: Substantial revisions of introduction should be performed to make the aims of the work and the experimental strategies more comprehensible to the readers.

Response 1: We agree with this comment. Therefore, we have revised and restructured the Introduction to make it more clear

Comment 2: Although the authors have reported the data about OCR, IHC and the level of secreted cytokines analyses, the corresponding techniques are not described in the manuscript. It is necessary to integrate them in the materials and methods section.

Response 2: Protocols were added to Materials and methods section

Comment 3: The Figure 4 pdf file is not attached to the manuscript and the captions should be included also in the supplementary figures. Furthermore, a list of acronyms should be included in the manuscript.

Response 3: Thank you for pointing this out. We have checked that all figures were attached to the manuscript and hope that the problem will not recur. The acronyms mentioned in text are now listed in the end of the article

Reviewer 3 Report

Comments and Suggestions for Authors

It’s a well-written study with interesting findings and lots of effort. I have following comments:

Please revise the title to present clear message of this study.

Please check English errors. 

Please explain why only female mice were used.

Please introduce MUC2-/- mutants in more details including other immune cells like DC or T cells in the intestine, and anything unique clinical symptoms such as skin, hair and lifespan etc., besides IBD. 

Please introduce the macrophages’ role in MUC2-/- mutants according to previous studies.

Please provide reference for the last sentence of Page 3/24:  “It is known that M1-activated macrophages generate ATP by glycolysis, which requires more oxygen consumption than energy production from oxidative phosphorylation, which is used by M0 and M2 polarized macrophages.” 

The IHC is impressive. Please provide the tunnel cell staining or alternative cell death staining of the MUC2-/- mutant intestine. 

Also, please provide evidence whether L-fucose can reduce apoptosis or necrosis of the MUC2-/- mutant intestine. 

The discussion and the conclusion are too long and not focused, which need to be combined. Please add subtitles in Discussion section, easy for readers. 

Please discuss the changes of intestine stem cell pool in MUC2-/- crypt;

Please discuss the changes of MUC gene family members in MUC2-/- mutants, if such information available. 

Please add scale bar in Supplementary Figure 1A and 1B.

Author Response

Dear reviewer,

We are grateful to you for your deep attention to our work and its high assessment.

Comment 1: Please revise the title to present clear message of this study.

Response 1: Thank you for your advise. We have, accordingly, revised the Title and now it looks like this: "Changes in the phenotype and metabolism of peritoneal macrophages in Mucin-2 knockout mice and partial restoration of their functions in vitro after L-fucose treatment"

Comment 2: Please check English errors.

Response 2: Done

Comment 3: Please explain why only female mice were used.

Response 3: Only females mice were used as male mice obtained from the same breeding were taken in other experiments, we added it to the Materials and methods section

Comment 4: Please introduce MUC2-/- mutants in more details including other immune cells like DC or T cells in the intestine, and anything unique clinical symptoms such as skin, hair and lifespan etc., besides IBD

Response 4: Starting at 16 weeks of age, Muc2-/- mice showed an increase in crypt length, loss of crypt architecture, and overall loss of intrinsic lamina structure in the proximal colon. In the distal colon, crypt hyperplasia, epithelial cell flattening and immune cell infiltration were observed as early as week 5. From the 12th to the 16th week of life, bocaloid cells are no longer detected in the distal part. At 16 weeks, there is a marked flattening and distortion of the epithelial structure, a large increase in crypt length, and active infiltration of inflammatory cells and epithelial erosion (Borisova et al., 2020; Van der Sluis et al., 2006). Histologic analysis of Mucin-2 gene knockout animals shows that bocaloid cells are almost undetectable in duodenal sections as well as in all sections of the mouse colon. The absence of normal mucosa in Mucin-2 knockout mice leads to the development of an inflammatory process with symptoms similar to those of patients with ulcerative colitis (Velcich et al., 2002). The main cytokines whose increased secretion is observed in patients with ILC are TNF-α, IL-1β and IL-6. In addition, the expression of IL-12 and INF-γ is altered. Similarly, increased expression of TNF-β and IL-1β was found in Muc2-/- mice. The expression of interleukin-1-alpha (IL-1α) was also increased in the mice compared to C57BL/6 mice, as was the expression of T regulatory cell transcription factor (Foxp3), whereas the expression levels of T helper-1 (Th1) and T helper-17 (Th17), Tbx21 and Rorc transcription factors were not different from the C57BL/6 control group. No differences were found in the expression of cytokines (IL-22, IL-17, IL-12p40, IL-12p70) involved in the T-helper mediated response (Borisova et al., 2020). The percent of CD103+CD11b− DCs was reduced in the colon lamina propria of Muc2-/-, while the percent of CD103+CD11b+ DC and CD103−CD11b+ macrophages remained unchanged (Wenzel et al., 2014). The number of CD45+CD19+, CD45+CD3+, CD45+CD3+CD4+, CD45+CD3+CD8+ cells in mesenteric lymph nodes of Muc2-/- mice was significantly higher than that of wild-type Muc2+/+ mice (Achasova et al., 2021).

Comment 5: Please introduce the macrophages’ role in MUC2-/- mutants according to previous studies

Response 5: 

In Muc2-/- role of the macrophages is tightly connected with chronic inflammation due to Mucin-2 absence. Declining of the intestinal mucous layer activates the immune system to pathogenic protection and to prevent hyperinflammatory. These changes are associated with the activation of antigen presenting cells (APCs), in particular tissue macrophages. In IBD, macrophages migrate massively through the intestinal mucosa and exhibit a phenotype and distribution distinct from tissue macrophages in homeostasis. In patients with Crohn's disease, macrophages also migrate through the muscle layer and mesenteric fat (Koch et al, 2010). In IBD, the disrupted epithelial barrier allows the contents of the intestinal lumen to invade the lamina propria, triggering an inflammatory response in the leukocytes of the lamina propria. As more than 200 genes have been associated with IBD, many of which are related to macrophage function, these immune cells represent a cell population that contributes to the pathogenesis of ulcerative colitis and Crohn's disease (Camacho-Pereira et al, 2016). Thus, CD103-CD11b+ cells, which are more abundant in the intestinal lumen of Muc2-/- mice, may resemble CD14+ macrophages, which are increased in the inflamed colon of patients with ulcerative colitis. This provides an opportunity to analyse the function of these cells in the development of colitis in an experimental model in Muc2-/- mice (Wenzel et al., 2014).

Comment 6: Please provide reference for the last sentence of Page 3/24:  “It is known that M1-activated macrophages generate ATP by glycolysis, which requires more oxygen consumption than energy production from oxidative phosphorylation, which is used by M0 and M2 polarized macrophages.”

Response 6:The reference was added, it was taken from Viola et al. 2019

Comment 7: The IHC is impressive. Please provide the tunnel cell staining or alternative cell death staining of the MUC2-/- mutant intestine.

Response 7: Tunel assay of Muc2-/- was already made in Velcich et al. and showed highly elevated ratio of proliferated to apoptic cells. Anna Velcich, et al., Colorectal Cancer in Mice Genetically Deficient in the Mucin Muc2. Science V295, 1726-1729 (2002). DOI: 10.1126/science.1069094

Comment 8: Also, please provide evidence whether L-fucose can reduce apoptosis or necrosis of the MUC2-/- mutant intestine.

Response 8: We did not mention that L-fucose has an effect on apoptosis or necrosis processes in the intestine but it might be an important part in further investigations. We are grateful for your idea. However in Muc2-/- mice Velcich et al. showed highly elevated ratio of proliferated to apoptic cells, which was added to the Introduction section of the article

Comment 9: The discussion and the conclusion are too long and not focused, which need to be combined. Please add subtitles in Discussion section, easy for readers.

Response 9: Thank you for pointing this out. We revised Discussion and Conclusion sections according on your recommendations to make them clearer and more logical.

Comment 10: Please discuss the changes of intestine stem cell pool in MUC2-/- crypt

Response 10: We did not investigate the role of stem cells in Muc2-/- IBD characterisation however it is considered that high proliferation levels demands higher numbers of epithelial stem cells (Binienda, 2020)

Comment 11: Please discuss the changes of MUC gene family members in MUC2-/- mutants, if such information available

Response 11: Sadly, there are no information available on this topic but it might be important in further studies to make sure we fully understand the role of mucins in IBD development

Comment 12: Please add scale bar in Supplementary Figure 1A and 1B

Response 12: Scale bars were added

Reviewer 4 Report

Comments and Suggestions for Authors

The manuscript by Arzhanova et al. entitled “Metabolism of colon and peritoneal macrophages during intestinal inflammation of Muc2-/- mice and L-fucose effect in vitro” investigates the impact of L-fucose on peritoneal macrophages. In general, the study is well designed and abundant experimental evidence is acquired. Figures are informative. However, several issues should be addressed.

Abstract: Abstract should be more focused on the findings of the current study.

Introduction:

-         Provide full names for Relmβ and iNOS, TLR, etc.

-         There are several spelling options in the text: MUC2, Mucin-2 or mucin2. Please unify it throughout the entire manuscript

-         Gene names should be italicized: e.g., Muc2 and Fuc2

-         The absence of fucose in the composition of mucin2 causes the development of IBD. This sentence should be rephrased. IBD is not caused by the fucose absence

-         The causes of different types of macrophage respiration are as-sociated with the accumulation of ROS and mitochondrial dysfunction. The reference and a detailed explanation should be provided

-         NAD: if it is an oxidized form, it should be written as NAD+

-         Please ensure the current use of subscripts: NAD+, Ca2+, etc.

Results:

-         Subheadings should be formulated as small conclusions.

-         Figure 2: scale bar is poorly visible

-         Occasionally ROS is used, sometimes reactive oxygen and nitrogen species. Use ROS and RNS in the entire manuscript

Materials and methods:

-         Why female mice were used? Provide the explanation in the text. Indicate their age and weight

-         Provide suppliers for DMEM, PBS, FBS, etc.

-          

-          

Comments on the Quality of English Language

Sometimes it is difficult to understand the meaning of sentences. 

Author Response

Dear Reviewer. 

We a grateful for your corrections, it will greatly improve the quality of the article. We have fixed the mistakes in the text. Please see the attachment

Round 2

Reviewer 1 Report

Comments and Suggestions for Authors

1.       About the abstract, it still only has descriptive language and no numerical data to support the results.

 2.       The Conclusion is too long, usually one paragraph is fine.

Author Response

Dear reviewer,

We are grateful to you for your deep attention to our manuscript and valuable comments, which allowed us to further improve it.

 Comment 1: About the abstract, it still only has descriptive language and no numerical data to support the results.

Response 1: Thank you for pointing this out. We have edited the Abstract and added the most significant numerical data to it.

Comment 2: The Conclusion is too long, usually one paragraph is fine.

Response 2: We agree with this comment. The Conclusion has been shortened to one paragraph.